# Frameworks for mitigating the risk of waterborne diarrheal diseases: A scoping review

**Chisala D. Meki**[1,2]*, **Esper J. Ncube**[2,3], **Kuku Voyi**[2]

**1** School of Public Health, University of Zambia, Lusaka, Zambia, **2** Faculty of Health Sciences, School of Health Systems and Public Health, University of Pretoria, Pretoria, South Africa, **3** Rand Water, Johannesburg, South Africa

* cdmeki@gmail.com

## Abstract

### Background

Diarrhea is one of the major cause of death and morbidity around the world.

### Objectives

This scoping review summarizes existing frameworks that aim to mitigate the risks of waterborne diarrheal diseases and describe the strengths and weaknesses of these frameworks.

### Eligibility criteria

Published frameworks designed to mitigate the risks of waterborne diarrheal diseases. Frameworks published in English, from around the world and published since inception to date.

### Sources of evidence

PubMed, Scopus, Web of Science, Google Scholar, Google Free Search, organization websites and reference lists of identified sources.

### Charting methods

Data were charted using the Joanna Briggs Institute tool. Results were summarized and described narratively. A criterion to score the strengths and weaknesses of the included frameworks was also developed.

### Results

Five frameworks were identified including: the hygiene improvement framework, community led total sanitation, global action plan for pneumonia and diarrhea, participatory hygiene and sanitation transformation, and sanitation and family education. These frameworks shared several common components, including identification of problems and risk factors, identification and implementation of interventions, and evaluation and monitoring. The frameworks

**Data Availability Statement:** All relevant data are within the manuscript under the list of references and Supporting information file(SI2).

**Funding:** This manuscript is part of a PhD study funded by the National Research Foundation South

Africa TWAS scholarship: Grant Numbers UID
116097 and UID 139119; URL https://www.nrf.ac.
za/ and the University of Zambia, Staff
Development Programme. The funders had no role
in the study design, data collection and analysis,
decision to publish, or preparation of the
manuscript.

**Competing interests:** The authors have declared
that no competing interests exist.

had several interventions including different infrastructure, health promotion and education, enabling environment and clinical treatments. Most of the frameworks included health promotion and education. All the frameworks were strengthened by including strategies for implementing and delivering intervention, human resource aspect, community involvement, monitoring, and evaluation. The main weakness included not having components for collecting, storing, and transferring electronic data and the frameworks not being specifically for mitigating waterborne diarrheal diseases. In addition, the identified frameworks were found to be effective in mitigating the risk of diarrhea diseases among other health effects.

## Conclusions

Existing frameworks should be updated specifically for mitigating waterborne diarrheal diseases that includes the strengths and addresses weaknesses of reviewed frameworks.

## Introduction

Diarrheal diseases are a major cause of morbidity and mortality around the world, especially in developing countries. The burden of diarrheal diseases is greatest in children under the age of five [1–3]. According to the World Health Organization (WHO), diarrheal diseases caused by unsafe drinking water, poor sanitation and hygiene have resulted in the deaths of an estimated 829,000 people [4,5]. The burden of diarrheal diseases is likely to grow because more than 2 billion people across the globe lack access to safely managed water services, safely managed sanitation services and basic services for handwashing [6].

Most diarrheal diseases are caused by waterborne pathogens that are ingested when people drink unsafe water that contains fecal matter. Waterborne diarrheal diseases include cholera, campylobacteriosis, typhoid fever, salmonellosis, shigellosis/ bacillary dysentery, cholera yersinosis, cryptosporidiosis, amebiasis (amoebic dysentery), cyclosporiasis and giardiasis and other gastroenteritis diseases caused by Adenoviruses, Astroviruses, Caliciviruses (e.g., Norwalk, Norwalk-like and Sapporo, Sapporo-like viruses), Enteroviruses (e.g., polio, echo, encephalitis, and Coxsackie viruses), Reovirus and Rotavirus [7–9]. The fecal-oral route plays an important role in understanding the transmission of diarrheal diseases [3]. Most available interventions that aim to prevent diarrheal diseases focus on halting the fecal-oral transmission route [10].

Many interventions are available to mitigate diarrheal diseases [11,12]. All these interventions can contribute to the holistic prevention of diarrheal diseases. These interventions include vaccines (e.g., rotavirus and measles), early and exclusive breastfeeding, vitamin A supplements, promoting handwashing with soap, improved water quantity and quality, household treatment and safe storage, providing sanitation services for solid and liquid waste management, health education and promotion. Many community clinics also have diarrhea treatment packages that include fluid replacement and zinc treatment [11,13,14]. To maximize effectiveness and long term sustainability, these interventions should be supported in legal and policy frameworks, have the necessary resources and involve stakeholders from the community, government, private sector and international communities [11,15].

Despite the availability and application of these interventions, waterborne diarrheal diseases are still recorded in developing countries [14]. In developing countries, the high incidence of waterborne diarrheal diseases may be due to various factors ranging from noncompliance to

interventions and interventions not being available where they are most needed [14,16]. Aside from individual interventions, operational frameworks and or approaches have been developed to reduce the risk of diarrheal diseases in communities. Examples of these frameworks include 'Community Led Total Sanitation' (CLTS), 'Participatory Hygiene and Sanitation Transformation' (PHAST) and the 'Hygiene Improvement Framework' (HIF) [17]. These frameworks consist of rules and ideas that aim to systematically deal with a particular problem, in this regard waterborne diarrheal diseases [18].

To date, no reviews have summarized the available frameworks for reducing the risk of waterborne diarrheal diseases. A scoping review of the available frameworks to reduce the risks of waterborne diarrheal diseases was conducted. Findings of this review may provide a platform for developing new frameworks or updating existing frameworks, which might ultimately help to attain the Sustainable Development Goals numbers three and six which addresses good health and wellbeing, and clean water and sanitation for all by 2030 [19].

### Objectives

The aim of this scoping review was to identify the frameworks for mitigating the risk of waterborne diarrhea diseases and critically review the frameworks to identify their strengths and weaknesses.

A scoping review was selected as it is important to identify and map existing literature as well as identify key concepts and gaps in research [20].

## Methodology

The Preferred reporting items for systematic reviews and meta-analysis for scoping reviews (PRISMA-ScR) was adhered to in this review [21] refer to S1 Table. The protocol for this scoping review was not published.

### Eligibility criteria

Data bases were searched with no limits on date of publication or setting. However, only frameworks reported in English were included due to lack of financial resources for translation. The review excluded proposals and only included final documents.

### Inclusion and exclusion criteria

Frameworks for mitigating the risk of diarrheal diseases were included. These included frameworks for preventing diarrhea, or frameworks for preventing and controlling or treating waterborne diarrheal diseases. The review considered the most recent versions of these frameworks and the frameworks had to be published in a reliable source. Frameworks that focused only on clinical treatment of diarrheal diseases, frameworks published in unreliable sources, frameworks that focused on animals and articles that we did not have access to were excluded. Further, studies that only reported interventions and mathematical models were also excluded. An eligibility criterion was created before literature search. Importantly, only frameworks that addressed diarrheal diseases in general were found. None of the existing frameworks specifically addressed waterborne diarrhea diseases. The list of included and excluded literature with reasons are presented in S1 File.

### Information sources

The following databases were searched: PubMed (13[th] April to 31[st] August 2021), Scopus (22[nd] April to 2[nd] August 2021) and Web of Science (22[nd] April to 2[nd] August 2021). Google Scholar

(23rd to 29th June 2021) and Google Free Search (2nd to 16th August 2021) were also searched. Further, websites of organizations including the WHO, United Nations Children's Fund (UNICEF), WaterAid, United States Agency for International Development (USAID), World Vision and the Foreign, Commonwealth and Development Office, World Bank and the Asian Development Bank were searched. Lastly, the reference lists of identified frameworks were also searched but found no additional frameworks. The search terms and full search strategies for PubMed, Scopus and Web of Science are presented in (S2 Table). The search terms were obtained from literature and refined in conjunction with a librarian at the University of Pretoria, South Africa and all the authors. The search strategy was also peer reviewed by independent researchers who have conducted similar reviews.

## Selection of articles

After the initial search, all the articles were downloaded into Endnote software where the duplicates were identified and removed. Two independent reviewers Chisala D. Meki and Esper J. Ncube (CDM and EJN) screened the titles and abstracts and selected the articles that presented suitable frameworks. A third reviewer Kuku Voyi (KV) acted as arbitrator to help resolve disputes.

## Data charting process

A data charting form was used to extract data from selected articles. The Joanna Briggs Institute (JBI) scoping review data extraction tool was modified to suit the review. The data charting form included the following key items:

1. Author(s)/developers

2. Year of publication

3. Origin/country of origin (where the source was published or conducted)

4. Aims/purpose

5. Components of framework

6. Intervention type, implementation areas, target groups

7. Effects of the framework/outcome measure

8. Outcome measure

The charting process was interactive, and the tool was modified as data were extracted. Two reviewers (CDM and EJN) charted the data, with a third reviewer acting as arbitrator (KV).

## Identifying strengths and weaknesses of frameworks

A scoring sheet to identify any strengths and weaknesses in the selected frameworks was developed. The scoring sheet was based on two existing frameworks, namely the Center for Disease Control and Prevention (CDC) framework for preventing or controlling communicable diseases [22] and the national framework for control of communicable diseases, Australia [23]. These frameworks were used as a benchmark because they both contain general components of frameworks. These two frameworks focus on general communicable diseases of which waterborne diarrheal diseases are a part.

In addition to the components obtained from the two standard frameworks, other components were added to the score sheet, including whether the framework identified and

quantified risk, whether the intervention targeted multiple groups, whether the intervention could be implemented within existing structures, whether the framework was sustainable and focused on waterborne diarrhea diseases [24–27] and included electronic means of data collection, storage and transfer. The 17 components included in the score sheet are presented in Table 1. The frameworks were independently scored by (CDM and EJN) with an independent arbitrator (KV).

## Critical appraisal of evidence

The included frameworks were not critically appraised, as they did not have specific study designs or outcome measures that can be measured using existing tools. To ensure quality, only frameworks published in reputable sources and by known organizations were included.

## Synthesis of results

The results of the reviews are presented in tables, and diagrams of the frameworks are also included. Each framework is also explained in a narrative synthesis. A Strengths, Weaknesses, Opportunities and Threats (SWOT) analysis was conducted to score the frameworks, using the scoresheet presented in Table 1.

## Ethical considerations

This study was approved by the Faculty of Health Science Research Ethics Committee of the University of Pretoria (REF: 847/2019) and the University of Zambia Biomedical Research Ethics Committee (UNZABREC) (REF: 808–2020). Informed consent was not considered in this study since no individual participants were included. The review only included already published literature.

# Results

## Selection of sources

Initially, a total of 9582 sources were retrieved from Scopus (n = 2641); Web of Science (n = 2575) and PubMed (n = 4227), and a further 139 sources from Google Scholar, Google Free Search, and organization websites. After removal of duplicates, 6246 documents were retained for title and abstract screening. A total of 6185 ineligible documents were removed, leaving 61 sources for full screening. After full screening, 56 sources were excluded for various reasons (Fig 1). Finally, five frameworks we identified and included for review.

## Study characteristics

Five eligible frameworks were identified namely: the hygiene improvement framework (HIF), community led total sanitation (CLTS), global action plan for pneumonia and diarrhea (GAPPD), participatory hygiene and sanitation transformation (PHAST) and the sanitation and family education (SAFE) framework. Since no frameworks that specifically addressed waterborne diarrheal diseases were found, frameworks that looked at diarrhea in general were included. The included frameworks were developed between the years of 1993 and 2009 and were developed by different organizations in different countries. The frameworks were developed to be implemented in different settings; one framework in rural areas, two frameworks in both rural and urban areas, and one framework at national level and another framework with no specified setting. Most of frameworks targeted communities and one framework targeted different groups including children, adults, households and societies (Table 2).

**Table 1. Score sheet used to assess the strengths and weaknesses of existing frameworks for mitigating the risk of diarrheal diseases.**

| | Criteria | | Framework |
|---|---|---|---|
| | Criterion and sources | Score | Definition |
| 1 | Problem identification [22,23] | 1 | Identify the problem using epidemiological and laboratory surveillance |
| | | 0.5 | Include only epidemiological surveillance or laboratory surveillance but not both |
| | | 0 | No problem identification |
| 2 | Risk identification and quantification (authors) | 1 | Has a component of risk identification and quantification |
| | | 0.5 | Included only risk identification or quantification |
| | | 0 | No component of risk identification and quantification |
| 3 | Identification of interventions [22,23] | 1 | Identification of intervention |
| | | 0 | No identification of interventions |
| 4 | Integrated approach [22] | 1 | Include at least hardware and software interventions |
| | | 0 | Includes either the software and hardware intervention(s) |
| 5 | Interventions target multiple groups (authors) | 1 | Intervention targets multiple groups in the community |
| | | 0 | Interventions only targets one group in the community e.g., children under 5 years only |
| 6 | Implementation and delivery of interventions [22] | 1 | Has component of implementation and delivery of intervention |
| | | 0 | No component of implementation and intervention delivery |
| 7 | Means of financing and or resource mobilization [22,23] | 1 | Has a component of means of financing or resource mobilization |
| | | 0 | No component of financing and or and resource mobilization |
| 8 | Human resources [23] | 1 | Component of required human resources in the program |
| | | 0 | No component of human resources required in the program |
| 9 | Implementation of interventions/program within existing structures (authors) | 1 | Intervention or programs implemented within existing structures |
| | | 0.5 | Not clear whether the program is implemented within existing structures but there is a component of implementation |
| | | 0 | No component on of implementation in existing structures |
| 10 | Multiple stakeholders' involvement [22,23] | 1 | Involvement of different stakeholders in diarrhea mitigation activities or program |
| | | 0.5 | Not too clear whether multiple stakeholders are involved in the program or activities |
| | | 0 | No involvement of multiple stakeholders |
| 11 | Community involvement [22,23] | 1 | Community involved in the whole process |
| | | 0.5 | Not clear whether there is community participation or not |
| | | 0 | No component of community involvement |
| 12 | Monitoring—follow-up [22,23] | 1 | Component of monitoring available |
| | | 0.5 | Not clear of monitoring component |
| | | 0 | Monitoring component not available |
| 13 | Evaluation—measure of success [22,23] | 1 | Evaluation component available |
| | | 0.5 | Not clear of availability of evaluation |
| | | 0 | No evaluation component |
| 14 | Electronic means of data collection, storage and transfer [22,23] (authors) | 1 | Availability of electronic means of data collection, storage and transfer |
| | | 0 | No means of electronic means of data collection, storage and transferring |
| 15 | Means of sustainability (authors) | 1 | Has a component of sustainability and explains the means of sustainability |
| | | 0.5 | Has a component of sustainability but means of sustainability not clearly explained |
| | | 0 | No component of sustainability |
| 16 | Focuses on waterborne diarrhea diseases (authors) | 1 | Framework focuses on waterborne diarrhea diseases |
| | | 0 | Framework does not focus on waterborne diarrheal diseases |
| 17 | Laws and policy development and improvement on intervention [22,23] | 1 | Component of laws and policy development or improvement |
| | | 0 | No component of laws and policy development or improvement |

Note: Authors—These components were included by the researchers.

Multiple stakeholder involvement: Program involving different government department or institutions, private sectors, non-governmental organizations, and international communities etc.

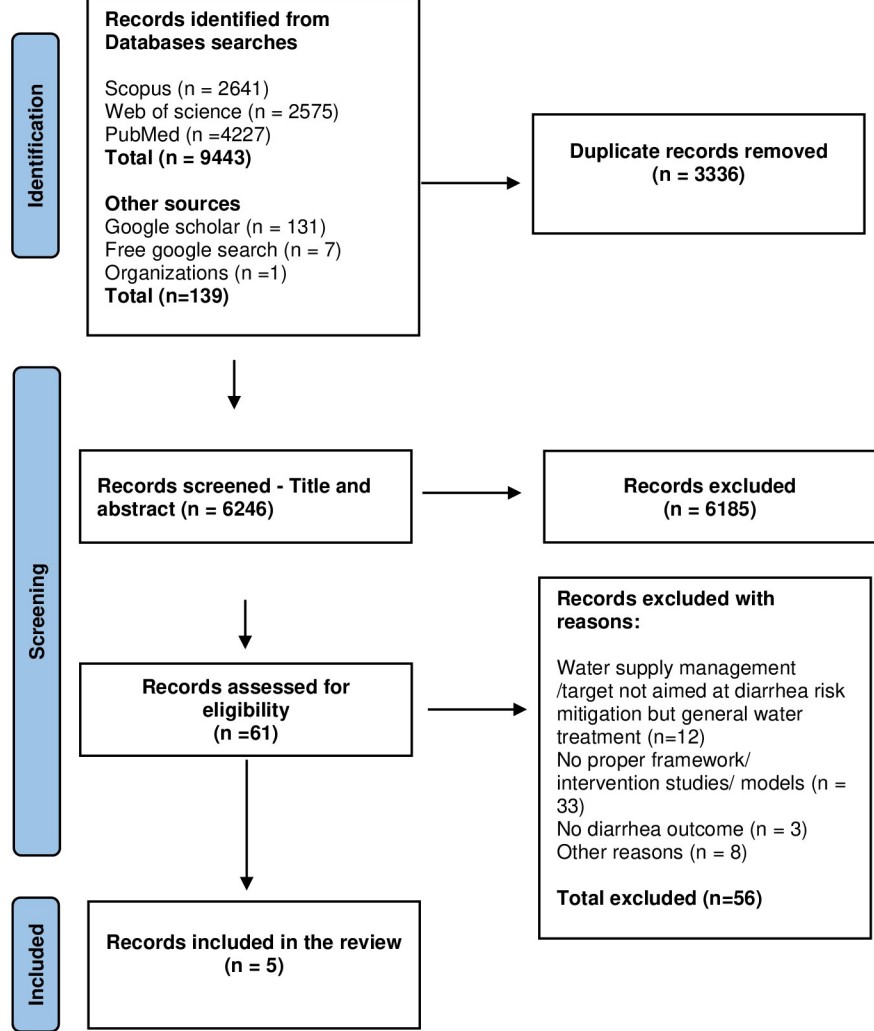

**Fig 1. PRISMA diagram for scoping review of frameworks for mitigating the risk of waterborne diarrheal diseases.**

## Description of frameworks

**Hygiene improvement framework.**   The hygiene improvement framework is a comprehensive framework for preventing diarrheal diseases that was created by the Environmental Health Program (EHP) and USAID in the USA. The framework has three facets, namely, health promotion, access to hardware and creating an enabling environment (Fig 2).

The hygiene promotion facet promotes hygiene by teaching and supporting behaviors that reduce diarrheal diseases in children and their caregivers. Household behaviors are encouraged including safe disposal of feces, washing hands correctly at the right times and, storing and using safe water for drinking and cooking. Preliminary community level studies should determine community members' knowledge about the causes of diarrhea, risk behaviors, enablers, and barriers to adopting appropriate behaviors. This will allow organizations to develop appropriate hygiene promotion interventions that will be accepted by the community. Hygiene promotion interventions include mass communication, social mobilization,

**Table 2. Characteristics of frameworks aimed at mitigating waterborne diarrheal diseases included in the scoping review.**

| | Name of framework | Sources | Year/ started | Implementation areas | Target population | Country / Organization |
|---|---|---|---|---|---|---|
| 1 | Hygiene improvement framework | Kleinau et al., and Environmental Health Project (EHP); UNICEF/Water [28,29] | 1999 | Rural and urban | Community, children, adults, households | EHP/USAID USA |
| 2 | Community led total sanitation | Kar and Chambers [30] | 1999 | Rural | Community | Bangladesh/ WaterAid |
| 3 | Global action plan for pneumonia and diarrhea | WHO/UNICEF [31] | 2009 | National level | National | WHO and UNICEF |
| 4 | Participatory hygiene and sanitation transformation | WHO [32] | 1993 | Rural and urban | Community | WHO/water and sanitation programs |
| 5 | The sanitation and family education | Bateman et al. [33] | 1995 | Not stated in the reference | Community | Bangladesh/CARE |

EHP: Environmental Health Program.

USA: United States of America.

community participation, social marketing and advocacy. Mass communication aims to increase awareness of hygiene facilities and good health practices through different channels including social media, music, dance, drama, literature, videos and home visits. Mass communication can happen at any community gathering, health facilities, learning institutions and households. In some settings, targeted training of health workers, teachers and community agents is an important communication strategy. Social mobilization and social marketing aim to involve all members of the community in disease control and hygiene promotion. To effectively promote hygiene behaviors, stakeholders, and civil societies, including governmental and non-governmental organizations, need to advocate for improved hygiene behaviors and interventions to support these behaviors. For example, providing hygiene education as well as water, sanitation and hand washing facilities for boys and girls in public schools may be a good entry point for sustainable hygiene improvement.

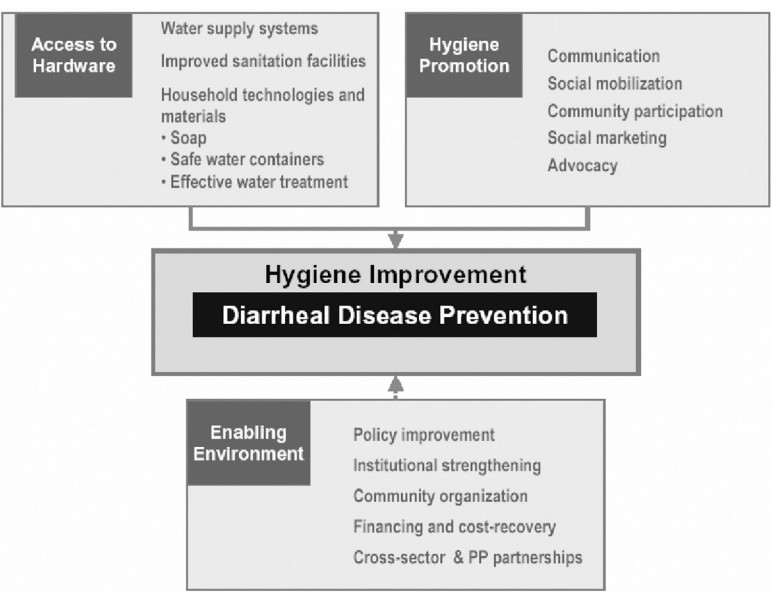

**Fig 2. The hygiene improvement framework [34].**

The second facet of the hygiene improvement framework is the hardware component. Hygiene can only be improved if there are adequate sanitation facilities for safe disposal of human waste. Adequate facilities such as latrines are needed to safely dispose of human excreta and avoid fecal contamination. The second hardware component necessary for good hygiene is adequate and good quality water supply. Household hygiene can also be improved by providing materials such as soap, safe water containers, household water treatment and potties for babies.

Hygiene promotion and having adequate hardware will only succeed if there is an enabling environment created by policy and legislative framework that improves hygiene, institutional governance, community participation, planning and financing, and private-public partnerships. An enabling environment will ensure the sustainability of any frameworks that are implemented. Even though the facets of the framework can be implemented individually, the hygiene promotion framework recommends that existing interventions be integrated into existing programs. The framework also recommends that the facets be implemented sequentially beginning with the hygiene promotion component for better outcomes with spillover effects on other diarrheal diseases related problems [28,29].

**Community led total sanitation.** Community-led total sanitation (CLTS) aims to prevent open defecation and keep rural communities free of open defecation. The community-led total sanitation approach was developed in Bangladesh by WaterAid with the aim of reducing open defecation in the communities. This approach raises awareness on the harms of open defecation to promote safe disposal of human waste. Communities are assisted to make ideal sanitary related decisions and attain their own sanitation solutions once they collectively decide to improve their sanitation practices. In the CLTS, communities must change their attitudes and behaviors, and adopt the use of community toilets. Community toilets were generally not used by community members and did very little to improve sanitation and hygiene, as well as prevent diseases.

The CLTS uses three stages to trigger collective behavioral change by encouraging and motivating people to confront the detrimental effect of open defecation. These stages are pretriggering, triggering and post-triggering. In the pretriggering stage, communities are selected, and facilitators are trained. Facilitators then collect baseline information and coordinate entry into the community. During the triggering stage, facilitators organize a community-wide meeting and conduct participatory exercises intended to trigger shame and disgust with open defecation. The first exercise is a 'walk of shame' where community members observe areas of open defecation. These areas are mapped, and volume of feces is calculated to quantify the amount of open feces lying in the area. The risk of disease transmission in specific areas is quantified using feces mobility mapping. It is likely that community participants will be more motivated towards improving their sanitary situation. In the post-triggering stage, routine follow-up visits are conducted to check the construction of latrines and extent of behavioral change. The open defecation free status of areas is verified, certified, and monitored [30,35].

**Global action plan for pneumonia and diarrhea.** Another framework, the GAPPD proposes a multi-sectoral, integrated approach to reducing morbidity and mortality due to pneumonia and diarrhea in children younger than five at national level by 2025. This framework was developed by the WHO and UNICEF. Pneumonia and diarrhea are the main causes of death and mortality among children globally. The GAPPD proposes an integrated framework of interventions proven to protect, prevent and treat childhood pneumonia and diarrhea in a coordinated way. Pneumonia and diarrhea both have similar determinants, preventive measures, and platforms to deliver interventions (Fig 3). The GAPPD was created primarily for national governments and their partners, and can also be used by global organizations, donor agencies and other organizations working on pneumonia and diarrhea. The GAPPD

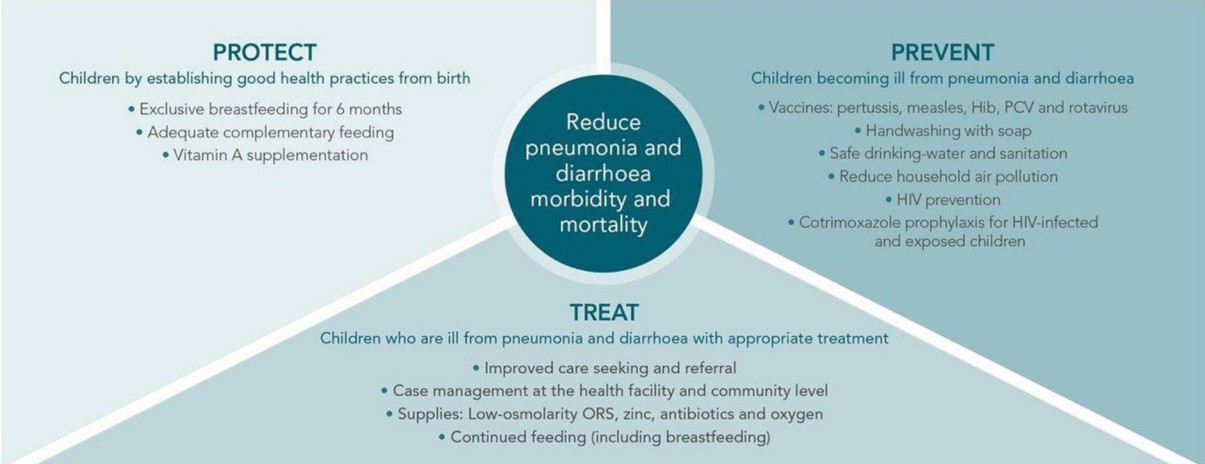

**Fig 3. GAPPD protect, prevent, and treat framework to reduce pneumonia and diarrhea [31].**

recognizes that strategies can only be implemented if communities and community members cooperate [31].

The GAPPD provides specific targets for each component. The protection component aims to ensure that 50% of children are exclusively breastfeed until they are six months old. The prevention component aims to achieve 90% immunization coverage of each of the following vaccines: pertussis, measles, Haemophilus influenzae type b (Hib), pneumococcal conjugate and rotavirus. For the treatment component, the GAPPD recommends that 90% of children with suspected pneumonia have access to treatment by an appropriate health care provider and access to antibiotics. Ninety percent of children with diarrhea should also have access to treatment with oral rehydration solution and zinc supplements. To save resources, the GAPPD recommends that these components should be implemented in existing healthcare services instead of working vertically.

The GAPPD recommends that governments address various components to achieve the specified goals:

- **Develop a clear country-level strategy and work plan, with assigned key responsibilities**: Generate political that will lead to responsive situation analysis for pneumonia and diarrhea and prioritize interventions. The strategy should include a costed medium to long term plan for accelerated action. Harmonization and collaboration between programs and sectors is critical to include private sector, academia, and civil society. Groups at greater risk or missed by services should be identified and targeted approaches should be implemented. Progress should also be monitored by developing a set of common indicators.

- **Coordinate implementation**: Establish a designated national working group for pneumonia and diarrhea prevention and control. This will help to mobilize resources, apply lessons from other integrated disease prevention and control efforts, track effective execution and evaluate systematic progress.

- **Engage and embed critical partners in the overall work plan/approach**: Involve other programs and sectors including the private sector, NGOs, United Nations agencies and other development cooperation partners.

- **Other actions**: Promote innovation, generate demand and ensure supply to overcome barriers to service delivery. Stakeholders should focus on implementing research and identifying optimal modes of delivery to reach those most in need [31,36].

 **Participatory hygiene and sanitation transformation (PHAST).** Participatory hygiene and sanitation transformation empowers communities to improve their hygiene behaviors, reduce diarrhea diseases and effectively manage water and sanitation. Community members are involved in planning and implementing interventions and hence experience a sense of ownership. Community members may say what they want or not in interventions. Community members are involved in monitoring and evaluating, which provides good feedback for improving activities. The PHAST comprises seven steps including problem identification, problem analysis, planning for solutions, selecting options, planning for monitoring and evaluation, and participatory evaluation (Fig 4). Each step is coupled with activities and tools that involve the community. The framework can be completed and implemented in about two weeks to six months. The PHAST should also be implemented sequentially to ensure best results [32,37].

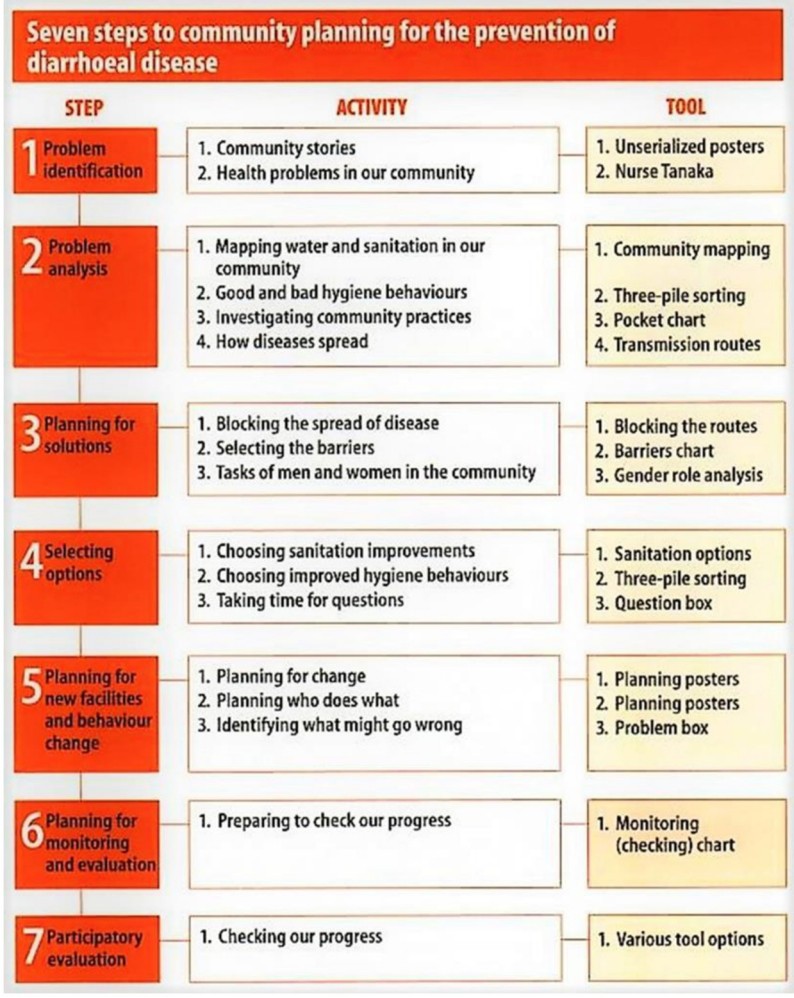

**Fig 4.  The PHAST framework for community planning to prevent diarrheal diseases [32].**

The PHAST works on the principles of involving facilitators to improve community awareness of water sanitation and hygiene through several activities. Through PHAST, community members formulate plans to improve sanitation by constructing and managing facilities as well as changing behaviors in the community and at individual level. The PHAST uses several tools including a picture series depicting local issues including water and sanitation. The PHAST requires that community members participate in workshops where they evaluate the local situation, identify problems and suggest solutions. Pocket charts are used to glean knowledge from participants and record their votes for possible solutions to ensure confidentiality. All the participants, including the facilitator, must be viewed as equals to ensure success [17,32].

**Sanitation and family education.**　The sanitation and family education (SAFE) approach promotes management of water, sanitation and hygiene using soft strategies. The SAFE approach followed on from the care water and sanitation/hygiene (WASH/CARE) project which was developed as a cyclone relief project in Bangladesh Chittagong in April 1991. The WASH/CARE project primarily rebuilt sanitation infrastructure including latrines, repaired damaged tube wells and built new tube wells. Following on from rebuilding, the SAFE project developed effective and replicable hygiene education strategies to promote behavior change, tested different models for health and hygiene education outreach, and designed and implemented a behavior-based monitoring system. The SAFE project had two outreach models. The first model focused on tube well caretakers, their spouses and tube well users. The second model involved outreach activities including school programs, child to child activities, and activities with key influencers in the community. These two models were compared to determine whether the more intensive outreach program would better influence hygiene behaviors. The SAFE program was implemented by facilitators in group discussions, demonstrations, participatory action learning exercises, flash card displays, folk songs, role playing, comic story sessions and games. These activities were designed and tested carefully to ensure relevance and appropriateness to local contexts.

The components of the SAFE included:

1. Hygiene education interventions based on information collected in small qualitative and quantitative research activities, rather than depending on stock messages and materials. Interventions reinforced existing positive behaviors or developed specific, appropriate alternatives to existing behaviors.

2. Hygiene behaviors were incrementally improved. Rather than promoting many ideal hygiene behaviors, SAFE identifies those behaviors most strongly associated with diarrhea in children and targets these priority behaviors with locally appropriate interventions.

3. Problems are identified using a behavior-based monitoring and improvement system. Community members are involved in analyzing problems and developing solutions. SAFE activities are continuously adjusted and improved.

4. Community members are encouraged to participate in every aspect of the project, including program design, outreach activities, monitoring and evaluation.

The SAFE approach is outlined in five steps (Fig 5). The first step determines goals and objectives. This is followed by developing a conceptual framework and outline of possible interventions. Specific behavioral interventions are then developed focusing on breaking the fecal-oral transmission route, including clean water, latrines and feces disposal, environmental cleanliness, hand washing, food hygiene and diarrhea management. Key problems are identified using baseline information on hygiene behavior and key areas for interventions. Baseline

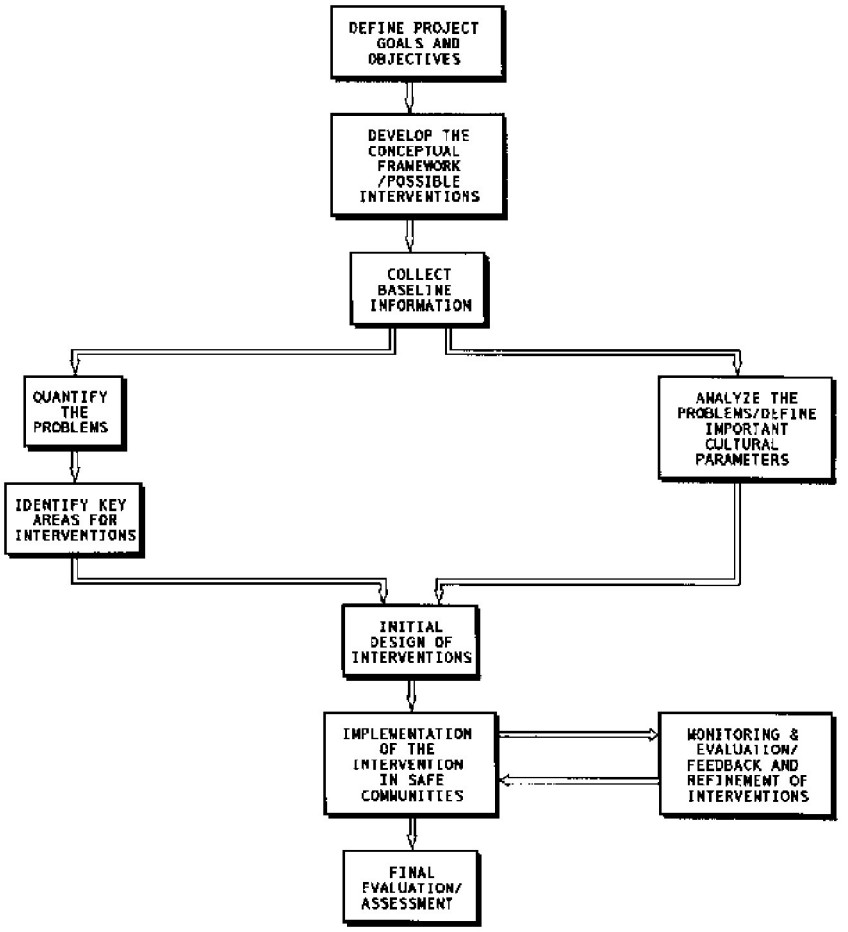

**Fig 5. The SAFE project life cycle.**

data are primarily collected using qualitative and quantitative studies. A monitoring system then measures key indicators for specific interventions. Key indicators are used to justify any necessary changes. The final surveys are essentially a repetition of the baseline surveys, and results are compared to evaluate effects of the SAFE intervention [33].

## Strengths and weaknesses of frameworks

The five frameworks included in this scoping review were scored (Table 3) using the 17 criteria outlined in Table 1. The following strengths were identified in all the frameworks. All the frameworks included facets describing implementation and delivery of interventions, human resources, community involvement, monitoring and evaluation. Strong frameworks also identified and quantified risks and involved all stakeholders. Frameworks were strengthened if they identified specific interventions and measures of sustainability. Three of the five frameworks targeted multiple groups and described financing and resource mobilization (Table 3). The frameworks scored half in problem identification and implementation of intervention or programs within existing structures (Table 3).

Three frameworks did not include suggestions for integrating laws and developing and improving policies. Similarly, three frameworks did not follow an integrated approach. None of the frameworks included electronic means of collecting, storing and transferring data. None

**Table 3. Strengths and weaknesses of frameworks for mitigating the risks of diarrheal diseases.**

| | Criteria /Component | Framework and Score | | | | | Total |
|---|---|---|---|---|---|---|---|
| | | HIF | CLTS | PHAST | GAPPD | SAFE | |
| 1 | Problem identification | Z(0.5) | Z(0.5) | Z(0.5) | Z(0.5) | Z(0.5) | 2.5 |
| 2 | Risk identification and quantification | Z(0.5) | Y(1) | Y(1) | Y (1) | Y(1) | 4.5 |
| 3 | Identification of interventions | N(0) | Y(1) | Y(1) | Y(1) | Y(1) | 4 |
| 4 | Interventions integrated approach | Y(1) | N(O) | N(0) | Y(1) | N(0) | 2 |
| 5 | Interventions target multiple groups | N (0) | Y(1) | Y(1) | N(0) | Y(1) | 3 |
| 6 | Implementation and delivery of interventions | Y (1) | Y(1) | Y(1) | Y(1) | Y(1) | 5 |
| 7 | Financing and or resource mobilization | Y(1) | N(0) | Y(1) | Y(1) | N(0) | 3 |
| 8 | Human resources | 1 | 1 | 1 | 1 | 1 | 5 |
| 9 | Implementation of interventions/program within existing structures | Y(1) | Z(0.5) | N(0) | Y(1) | N(0) | 2.5 |
| 10 | Multiple stakeholders' involvement | Y(1) | Y(1) | Y(1) | Y(1) | Z(0.5) | 4.5 |
| 11 | Community involvement | Y(1) | Y(1) | Y(1) | Y(1) | Y(1) | 5 |
| 12 | Monitoring–follow up | Y(1) | Y(1) | Y(1) | Y(1) | Y(1) | 5 |
| 13 | Evaluation—measures of success | Y(1) | Y(1) | Y(1) | Y(1) | Y(1) | 5 |
| 14 | Electronic means of data collection, storage and transferring | N(0) | N(0) | N(0) | Z(0) | N(0) | 0 |
| 15 | Means of sustainability | Y(1) | Y(1) | Z(0.5) | Y(1) | Z(0.5) | 4 |
| 16 | Focuses on waterborne diarrhea diseases | N(0) | N(0) | N(0) | N(0) | N(0) | 0 |
| 17 | Laws and policy development and improvement on intervention | Y(1) | N(0) | N(0) | Y(1) | N(0) | 2 |
| | Total | 12 | 10 | 11 | 12.5 | 9.5 | |

N = Not available = 0; Y = available = 1; Z = Available but not too clear or adequate = 0.5.

of the frameworks focused specifically on mitigating waterborne diarrheal diseases (Table 3). None of the frameworks scored 17/17, but all the frameworks scored at least 9.5/17. The GAPPD framework scored the highest with 12.5/17 and SAFE scored the lowest (9.5/17).

In addition to the 17 components assessed in the criteria, the reported effectiveness of the identified frameworks was assessed. Positive findings have been revealed in implementation of the frameworks in different places world over. An Assessments of the effectiveness of CLTS has revealed increase in construction of latrines and use as well as reduction in diarrhea cases in areas of implementation in Ethiopia, Ghana and Uganda [17,38–41]. However, evidence from Mali showed no difference in diarrhea cases between CLTS implemented and non-implemented areas [42]. In terms of PHAST, no study was found that reported on the effectiveness of the approach to prevent diarrhea diseases. However, some assessments have revealed effectiveness of the approach in the promotion of sustainable hygiene behaviors change, improved sanitation and conveying health messages [17,43,44]. In the case of GAPPD, deaths due to Pneumonia and diarrhea have fallen by 27% globally since the framework was introduced despite low- and middle-income countries still lagging in achievement of the GAPPD goals [45]. Implementation of the HIF framework in Guatemala revealed a reduction in diarrhea diseases among under-fives children by 4.5% through hand washing interventions that was implemented within the HIF framework [28,46]. Lastly, implementation of SAFE approach in Chittagong Bangladesh revealed a two third reduction in the cases of diarrhea in safe interventions groups compared to the controls [47].

## Discussion

In this scoping review, five frameworks that aimed at mitigating the risk of diarrheal diseases were included. These frameworks were developed by different organizations, implemented at

different levels, and all aimed to prevent diarrhea especially among children. All these frameworks had components of community involvement, problem and risk factor identification, identification and implementation of interventions and methods of evaluating and monitoring, which were regarded as strengths. Most of the frameworks recommended that hygiene could be improved by providing hardware, including sanitation infrastructure. Soft interventions included health promotion and creating an enabling environment for changing behavior. The frameworks had several strengths and weaknesses. One of the key weaknesses was that none of the frameworks included strategies for collection, storage, and transfer of electronic data. In addition, none of the frameworks focused specifically on waterborne diarrhea diseases. The identified gaps were seen as the major weaknesses of the frameworks.

Notably, all the frameworks in this scoping review contained a component of problem identification even though none of the frameworks used epidemiological assessments and laboratory tests to determine the cause of diarrheal diseases. Laboratory tests are important for identifying specific pathogens which will then determine the specific mitigation measures [48]. For example, chlorine, which can be used to destroy most diarrheal diseases pathogens, cannot destroy cryptosporidium parasites which cause cryptosporidiosis, a type of diarrhea disease [49]. All the frameworks also identified and quantified risk factors, and appropriate interventions. This is important as each community usually has unique risk factors for diarrheal diseases which need to be addressed with specific interventions to be effective [26].

Most of the frameworks targeted multiple groups in the community, which ensures that all people in the community are reached, and uptake is maximized [50]. Even though most interventions aim to reduce diarrhea in children, interventions must benefit all community members since adults are also affected by diarrhea [51,52]. Community involvement will also increase ownership which is integral to the sustainability of health programs [53,54]. Most of the frameworks in this review addressed sustainability which is an important aspect of health programs. Without sustainability plans, programs may waste resources. Diarrhea interventions should also be continuous to prevent reoccurrence of disease [24].

All the frameworks had components of monitoring and evaluation which are important aspect of sustainability. Sanitation infrastructure and water quality should be continuously monitored to ensure that everything is working well and to check for the effectiveness of interventions to help improve implementation when needed [55,56]. The majority of the frameworks also addressed the availability of human resources, which are required at almost all levels of health program implementation [57]. The availability of human resources also ties in with the involvement of multiple stakeholders. All the frameworks required the involvement of multiple stakeholders. Improved hygiene requires different players to be involved including government institutions such as local authorities and health departments, as well as the private sector and international communities [58]. Involving multiple stakeholders will facilitate financing and resource mobilization [58,59].

In this review, some of the frameworks lacked certain important components for mitigating diarrhea diseases. For example, not all the frameworks were designed to be implemented in existing programs, nor did they describe how they should be integrated into laws and policies. Few frameworks described an integrated approach such as using multiple interventions to prevent diarrheal diseases. When interventions are implemented in existing programs, sustainability and efficient use of resources will be promoted. This is less expensive than designing new programs from scratch and running vertical health programs [27]. Frameworks should describe how they can be implemented in laws and policies to ensure that standards are maintained [59]. For example, laws should dictate which type of toilet should be built to avoid ground water contamination in specific areas. Frameworks should also allow for an integrated approach depending on the needs of the community. Some communities may require better

sanitation infrastructure and water treatment services, while other communities may need to be educated about better hygiene behavior and the importance of vaccinations [60,61]. None of the frameworks included a component for collecting, storing and transferring electronic data. Electronic collection, storage and transfer of data eases data management and facilitates easy decision making [59]. Another weakness was that none of the frameworks specifically addressed waterborne diarrheal diseases.

Of the frameworks reviewed in this study, particularly the GAPPD framework addressed most of the components included in scoring sheet used in this review. Although the framework scored the highest, the GAPPD did not target multiple groups as it only targeted children under the age of five. In addition, it had no means of electronic data collection, storage and transfer, and did not specifically focus on waterborne diarrheal diseases. This framework probably included the most favorable components because it is a global plan to reduce diarrhea and pneumonia mortality and morbidity. After the GAPPD, the HIF framework included the most components followed by PHAST then CLTS. The SAFE framework included the fewest desirable components.

## Proposed framework

Based on the findings of this review, this study proposes the use of the various components in the identified frameworks as basis for the development of future frameworks. This is in consideration that the identified frameworks were found to be effective in mitigating the risk of diarrhea diseases among other health benefits [17,28,38–44,46,47]. The proposed framework consists of six (6) components which should be implemented in series:

The first component is **problem identification** involving community surveillance must be conducted to determine the burden, disease causative agent [62], high risk areas, characteristics of cases and other related factors of waterborne diarrhea diseases. The second component is **Identification and quantifying of risks** factors in the community should be done focusing on water supply, sanitation, and hygiene (hand washing) the major risk factors of waterborne diarrhea diseases [61,63] and other demographic, social economic behavioral and environmental related factors [64]. This should be followed by **identification of evidence-based intervention(s)** based on the risk's factors identified. Use of multiple interventions is proposed since diarrhea has multiple risk factors [8,31]. Prioritizing interventions based on the needs of the community and provision of long terms investments in water supply and toilets is proposed as these are fundamentals of diarrhea prevention [46]. Then **assessment of intervention(s) in target community** involving getting information on past and current interventions from the target communities and other relevant stakeholders must be done. This information will be used as basis of deciding on the required and best way of implementation of the interventions. **Selection and adoption of intervention(s)** of the diseases should then be done based on the risks, interventions identified as well as the assessment of intervention in the community. This selection should be based on appropriateness and acceptability of the interventions through evidence-based literature, experts' knowledge and community involvement.

The last component involves **implementing the selected intervention(s)** in the community. This should be followed by **monitoring** and **evaluation** of the implementation involving checking the progress of the implementation and effect of the implemented intervention(s). Means of **sustainability** of the intervention(s) must also be considered to avoid reoccurrence of diseases. Sustainability should be ensured by involving the community in all the stages of the framework to encourage ownership [53,54].

In addition to the six components, **system support factors** have been identified including: Intersectoral participations; government will; human and financial resources and resource

mobilization; policies and laws; strengthening collection and recording of data through electronic means; adapting the components to emerging problems and new solutions; working within available structures: horizontal approach of programming; institutional strengthening; provision of laboratory facilities for testing and development or strengthening of a national preparedness program for waterborne diarrheal diseases outbreaks [28,31]. A country level framework for use in mitigating the risk of waterborne diarrhea diseases in peri-urban areas of Zambia and similar settings will be developed based on the proposed components.

## Limitations

Databases that were not in English were not searched, but a variety of databases were searched. Gray literature was also searched. The literature searches were also done over a limited six-month period but all literature from inception to date were included. Quality checks were not conducted on the included frameworks, but only included frameworks that were created by reputable organizations and obtained from reputable websites or databases were included. It is also important to note that one of the components that was used to assess the strengths and weaknesses of the framework's availability of an electronic means of data collection, storage and transferring, may not necessarily be a weakness for most of the frameworks as they were created before advancement in electronic means. Nevertheless, it was included as it is applicable to the current frameworks and is an important component for inclusion in future frameworks. Some of the frameworks reviewed were clearly not frameworks but rather approaches thus their low rating. These plans or approaches still provided some important information to mitigate the risk of waterborne diarrhea diseases and can be used to develop and improve the framework.

## Conclusion

This study reviewed frameworks for mitigating the risk of diarrheal diseases. No frameworks specifically addressing waterborne diarrheal diseases were found thus only frameworks aiming to mitigate diarrheal diseases in general were included. Further, 17 favorable components that could be included in future frameworks were identified. Most of the frameworks in this review had the favorable components of identifying problems and risk factors, identifying and implementing interventions and evaluating and monitoring outcomes. The interventions ranged from improving sanitation infrastructure and water quality to hygiene promotion and education, whilst creating an enabling environment. None of the frameworks included an element on collecting, storing or transferring electronic data, or focused specifically on waterborne diarrhea diseases. The identified frameworks were found to be effective in mitigating the risks of diarrhea among other health effects. Based on these results, this review has proposed a framework that will consist of six components including: problem identification; identification and quantifying risk factors; identification of evidence-based interventions; assessment of interventions in target communities; selection and adoption of interventions and implementing, monitoring and evaluation and means of sustainability of the interventions. Lastly, system support factors of these components must also be considered to mitigate the risk of waterborne diarrhea diseases.

## Supporting information

**S1 Table. PRISMA-ScR checklist.**
(DOCX)

**S2 Table. Complete search strategy PubMed, Scopus and Web of Science.**
(DOCX)

**S1 File. Included and excluded literature with reasons.**
(DOCX)

## Acknowledgments

Dr. Cheryl Tosh for editing.

## Author Contributions

**Conceptualization:** Chisala D. Meki, Esper J. Ncube.

**Data curation:** Chisala D. Meki.

**Formal analysis:** Chisala D. Meki, Esper J. Ncube, Kuku Voyi.

**Methodology:** Chisala D. Meki, Esper J. Ncube, Kuku Voyi.

**Project administration:** Chisala D. Meki.

**Supervision:** Esper J. Ncube, Kuku Voyi.

**Validation:** Chisala D. Meki, Esper J. Ncube, Kuku Voyi.

**Visualization:** Chisala D. Meki.

**Writing – original draft:** Chisala D. Meki.

**Writing – review & editing:** Chisala D. Meki, Esper J. Ncube, Kuku Voyi.

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
