## [Decision Letter · Decision Letter 0]

15 Jul 2022

PONE-D-22-14568Frameworks for mitigating the risk of waterborne diarrheal diseases: A scoping reviewPLOS ONE

Dear Dr. MEKI,

Thank you for submitting your manuscript to PLOS ONE. After careful consideration, we feel that it has merit but does not fully meet PLOS ONE’s publication criteria as it currently stands. Therefore, we invite you to submit a revised version of the manuscript that addresses the points raised during the review process.

Please see below the comments and suggested MAJOR revisions made by the individual(s) who reviewed your manuscript.  If provided, the referee's report(s) indicate the revisions that need to be made before it can be accepted for publication.

We look forward to receiving your revised manuscript.

Kind regards,

Ricardo Santos

Academic Editor

PLOS ONE

Journal Requirements:

4. Please ensure that you include a title page within your main document. You should list all authors and all affiliations as per our author instructions and clearly indicate the corresponding author.

Reviewers' comments:

Reviewer's Responses to Questions

**Comments to the Author**

1. Is the manuscript technically sound, and do the data support the conclusions?

Reviewer #1: Partly

Reviewer #2: Yes

2. Has the statistical analysis been performed appropriately and rigorously? 

Reviewer #1: N/A

Reviewer #2: No

3. Have the authors made all data underlying the findings in their manuscript fully available?

Reviewer #1: Yes

Reviewer #2: Yes

4. Is the manuscript presented in an intelligible fashion and written in standard English?

Reviewer #1: No

Reviewer #2: Yes

5. Review Comments to the Author

Reviewer #1: lines 18 and 19: health promotion and

19 education, and enabling environment and clinical treatments

Comment: and must not be repeated

Line 37 to 39

Waterborne diarrheal diseases include cholera,

campylobacteriosis, typhoid fever, salmonellosis, shigellosis/ bacillary dysentery, cholera

yersinosis, cryptosporidiosis, amebiasis (amoebic dysentery), cyclosporiasis and giardiasis

Comment: where are viral causes

line 74:This scoping review summarizes the

Comment: This sentence needs to be re-written

Line 78: Scoping reviews are appropriate to identify and map existing evidence, identify key

Comment: This sentence needs to re-written

Line 125: We used a data charting form to extract data from selected articles. We modified the

Comment: to writ we......., we.....

This type of writing needs to be changed

Line 251: . The approach was developed in Bangladesh by WaterAid.

Comment: needs more details

Line 298: • Develop a clear country-level strategy and work plan, with assigned key responsibilities:

Comment: Please, write the same issue in the manuscript using the same manner

Lines 387 and 388: We scored the five frameworks included in this scoping review (Table 4) using the 17

criteria outlined in Table 2. We identified the following strengths in all the frameworks. All the

Comment: in all parts of the manuscript, english needs to be improved which We.....we....we

The use of We in the first of all sentences needs to be changed.

Lines 485 to 488:

Conclusion

We reviewed frameworks for mitigating the risk of diarrheal diseases. No framework

specifically addressed waterborne diarrheal diseases thus we included frameworks aiming to

mitigate diarrheal diseases in general.

Comment: Please, write the conclusions directly.

English language in all the manuscript needs to be revised and the method of writing needs to be revised. The structure of the manuscript and the method authors expressed the data needs to be revised.

Reviewer #2: Authors have summarized frameworks related to mitigation of diarrheal diseases, focusing on waterborne diarrhoea.

It may also be important to analyse/ incorporate the outcomes of implementing these frameworks in the evaluation scoring system, if such data are available. It may also provide insights on the weaknesses of these frameworks.

Authors have provided very few suggestions for developing future frameworks to minimize the waterborne diarrhoea (line 450-465). This section needs to be expanded using examples from the selected frameworks, other studies, and author’s own opinions.

Authors have highlighted the unavailability of electronic data collection in the manuscript several times. However, several studies were conducted in a time where electronic devices are not as common as today and digital literacy is not really high. Therefore, it may not be directly highlighted as a weakness.

6. PLOS authors have the option to publish the peer review history of their article (what does this mean?). If published, this will include your full peer review and any attached files.

Reviewer #1: **Yes: **Waled Morsy El-Senous

Reviewer #2: No

---

## [Author Response · Author response to Decision Letter 0]

7 Sep 2022

All the Journal requirements have been addressed as requested. 

The response to the reviewers have also be addressed as presented below: 

REVIEWER 1 

 1. lines 18 and 19: health promotion and

19 education, and enabling environment and clinical treatments

Comment: and must not be repeated 

Response: Thank you for the feedback. One ‘and’ has been removed line 45

2.line 37 to 39 Waterborne diarrheal diseases include cholera,

campylobacteriosis, typhoid fever, salmonellosis, shigellosis/ bacillary dysentery, cholera

yersinosis, cryptosporidiosis, amebiasis (amoebic dysentery), cyclosporiasis and giardiasis

Comment: where are viral causes 

Response: Thank you for this feedback. The viral causes have been added as advised lines 67 to 70

3. line 74: This scoping review summarizes the

Comment: This sentence needs to be re-written 

Response: Thank you. The objective of the review has been rewritten. Lines 106 to 108.

4. Line 78: Scoping reviews are appropriate to identify and map existing evidence, identify key

Comment: This sentence needs to re-written 

Response: Thank you for the comment. The sentence has been rewritten. Lines 112 to 113 

5. Line 125: We used a data charting form to extract data from selected articles. We modified the

Comment: to writ we......., we.....

This type of writing needs to be changed 

Response: Revised thank you. Lines 166 to 168 

6. Line 251: . The approach was developed in Bangladesh by WaterAid.

Comment: needs more details 

Response: Thanks. Details have been included. Lines 303 to 305 

7. Line 298: • Develop a clear country-level strategy and work plan, with assigned key responsibilities:

Comment: Please, write the same issue in the manuscript using the same manner 

Response: Thank you for this commnet. The proposed framework is developed for use in Zambia and similar set up as indicated in the proposed framework lines 588 to 590

8. Lines 387 and 388: We scored the five frameworks included in this scoping review (Table 4) using the 17

criteria outlined in Table 2. We identified the following strengths in all the frameworks. All the

Comment: in all parts of the manuscript, english needs to be improved which We.....we....we

The use of We in the first of all sentences needs to be changed. 

Response: The language in all parts of the manuscript has been revised. Please kindly check the whole manuscript. 

9. Lines 485 to 488: Conclusion

We reviewed frameworks for mitigating the risk of diarrheal diseases. No framework

specifically addressed waterborne diarrheal diseases thus we included frameworks aiming to

mitigate diarrheal diseases in general.

Comment: Please, write the conclusions directly.

English language in all the manuscript needs to be revised and the method of writing needs to be revised. The structure of the manuscript and the method authors expressed the data needs to be revised. 

Response: Thank you for this comment. The conclusion has been modified as recommended check lines 610 to 627. Kindly note that the whole manuscript has been edited. 

REVIEWER 2 

1. Authors have summarized frameworks related to mitigation of diarrheal diseases, focusing on waterborne diarrhoea.

It may also be important to analyse/ incorporate the outcomes of implementing these frameworks in the evaluation scoring system, if such data are available. It may also provide insights on the weaknesses of these frameworks. 

Response: Thank you for this recommendation. The outcomes for the implementation of frameworks have been added in the results section not necessarily in the scoring system but in a descriptive way please check lines 462 to 478 we have also added related information in the discussion. 

2. Authors have provided very few suggestions for developing future frameworks to minimize the waterborne diarrhoea (line 450-465). This section needs to be expanded using examples from the selected frameworks, other studies, and author’s own opinions. 

Response: Thank you for this comment. More information of the future framework have been provided lines 551 to 590 

3. Authors have highlighted the unavailability of electronic data collection in the manuscript several times. However, several studies were conducted in a time where electronic devices are not as common as today and digital literacy is not really high. Therefore, it may not be directly highlighted as a weakness. 

Response: Thank you for this valuable comment. We have not removed the electronic means of data collection, tranfers and storage in the scoring criteria. However, we have included it under the limitation section considering the comment given. Check lines 598 to 603

---

## [Decision Letter · Decision Letter 1]

14 Nov 2022

Frameworks for mitigating the risk of waterborne diarrheal diseases: A scoping review

PONE-D-22-14568R1

Dear Dr. MEKI,

We’re pleased to inform you that your manuscript has been judged scientifically suitable for publication and will be formally accepted for publication once it meets all outstanding technical requirements.

Kind regards,

Ricardo Santos

Academic Editor

PLOS ONE

Additional Editor Comments (optional):

Reviewers' comments:

Reviewer's Responses to Questions

**Comments to the Author**

1. If the authors have adequately addressed your comments raised in a previous round of review and you feel that this manuscript is now acceptable for publication, you may indicate that here to bypass the “Comments to the Author” section, enter your conflict of interest statement in the “Confidential to Editor” section, and submit your "Accept" recommendation.

Reviewer #2: All comments have been addressed

Reviewer #3: All comments have been addressed

2. Is the manuscript technically sound, and do the data support the conclusions?

Reviewer #2: Yes

Reviewer #3: Yes

3. Has the statistical analysis been performed appropriately and rigorously? 

Reviewer #2: No

Reviewer #3: Yes

4. Have the authors made all data underlying the findings in their manuscript fully available?

Reviewer #2: Yes

Reviewer #3: Yes

5. Is the manuscript presented in an intelligible fashion and written in standard English?

Reviewer #2: Yes

Reviewer #3: Yes

6. Review Comments to the Author

Reviewer #2: Line 69-70: Norovirus and Sapovirus will be enough

Line 122: Databases

Line 504, 537: Diarrheal diseases

Line 546: involve

Line 535-558: The bold coloured words and the other words may not exactly connect grammatically. Please check.

Reviewer #3: The authors addressed the previous comments to primary reviewers. This is an important topic for prevention of waterborne diseases.

7. PLOS authors have the option to publish the peer review history of their article (what does this mean?). If published, this will include your full peer review and any attached files.

Reviewer #2: No

Reviewer #3: No

---

## [Editor Report · Acceptance letter]

23 Nov 2022

PONE-D-22-14568R1 

Frameworks for mitigating the risk of waterborne diarrheal diseases: A scoping review 

Dear Dr. Meki:

I'm pleased to inform you that your manuscript has been deemed suitable for publication in PLOS ONE. Congratulations! Your manuscript is now with our production department. 

Kind regards, 

on behalf of

Dr. Ricardo Santos 

Academic Editor

PLOS ONE